# Quantifying the uncertainties in thermal–optical analysis of carbonaceous aircraft engine emissions: an interlaboratory study

**Timothy A. Sipkens**[1], **Joel C. Corbin**[1], **Brett Smith**[1], **Stéphanie Gagné**[1], **Prem Lobo**[1,a], **Benjamin T. Brem**[2,3], **Mark P. Johnson**[4], **and Gregory J. Smallwood**[1]

[1]Metrology Research Centre, National Research Council Canada, TS1 Ottawa, Canada
[2]Empa, Swiss Federal Laboratories for Materials Science and Technology, TS2 Dübendorf, Switzerland
[3]Laboratory of Atmospheric Chemistry, Paul Scherrer Institute, TS3 Villigen, Switzerland
[4]Rolls-Royce plc, Derby, UK
[a]current address: Office of Environment and Energy, Federal Aviation Administration, Washington, DC 20591, USA

**Correspondence:** Timothy A. Sipkens (timothy.sipkens@nrc-cnrc.gc.ca)

**Abstract.** CE1 TS4 Carbonaceous particles, such as soot, make up a notable fraction of atmospheric particulate matter and contribute substantially to anthropogenic climate forcing, air pollution, and human health impacts. Thermal–optical analysis (TOA) is one of the most widespread methods used to speciate carbonaceous particles and divides total carbon (TC) into the operationally defined quantities of organic carbon (OC; carbon that has evolved during slow heating in an inert atmosphere) and elemental carbon (EC). While multiple studies have identified fundamental scientific reasons for uncertainty in distinguishing OC and EC, far fewer studies have reported on between-laboratory reproducibility. Moreover, existing reproducibility studies have focused on complex atmospheric samples. The real-time instruments used for regulatory measurements of the mass concentration of aircraft engine non-volatile particulate matter (nvPM) emissions are required to be calibrated to the mass of EC, as determined by TOA of the filter-sampled emissions of a diffusion flame combustion aerosol source (DF-CAS). However, significant differences have been observed in the calibration factor for the same instrument based on EC content determined by different calibration laboratories. Here, we report on the reproducibility of TC, EC, and OC quantified using the same TOA protocol, instrument model (Model 5L, Sunset Laboratory), and software settings (auto-split-point: Calc405) across five different laboratories and instrument operators. Six unique data sets were obtained, with one laboratory operating two instruments. All samples were collected downstream of an aircraft engine after treatment with a catalytic stripper to remove volatile organics. Between-laboratory contributions made up a majority of the within-filter uncertainties for EC and TC, even for these relatively well-controlled samples. Overall, expanded ($k = 2$) uncertainties due to measurement reproducibility correspond to 17 %, 8.0 %, and 12 % of the nominal values for EC, OC, and TC, respectively, and 6.8 % in the EC / TC ratio. These values are lower than previous studies, including atmospheric samples without volatile organic removal; therefore, they likely represent lower limits for the uncertainties of the TOA method.

## 1 Introduction

Carbonaceous particles contribute to both natural and anthropogenic climate forcing, air pollution, and human health impacts. The aviation industry remains a notable source of these particles, and it will continue to be important as air transportation continues to expand CE2 . Unlike $CO_2$, particulate matter (PM) emissions from the aviation industry contain larger uncertainties, as does their effect on contrails and cloud formation (Righi et al., 2021). For aircraft engine emissions, thermal–optical analysis (TOA) is currently the reference standard for calibrating the instruments used to measure the mass concentration of non-volatile particulate matter (nvPM) emitted by aircraft engines (SAE, 2018; Lobo et

al., 2015a, 2020). However, open questions remain with respect to the uncertainties and associated metrology (referring to the establishment of uncertainties by way of interlaboratory comparisons to the traceability) of these measurements.

In TOA, the total carbon (TC) mass collected on a quartz filter is measured in two distinct phases. First, the total carbon mass that has evolved from a sample during controlled heating in an inert environment is considered organic carbon (OC), while the remainder, heated in an oxidizing environment and corrected for pyrolysis, is considered elemental carbon (EC) (Birch and Cary, 1996). If the mass fraction of carbon in OC (40 %–80 %; Turpin and Lim, 2001; Bae et al., 2006) or in EC (90 %–98 %; Figueiredo et al., 1999; Singh and Vander Wal, 2020; Corbin et al., 2020) is known, these quantities can then be used to estimate the total mass of carbonaceous particles on the filter.

It is well known that the widely variable properties of carbonaceous materials lead to significant uncertainties in the separation of TC into OC and EC (Watson et al., 2005; Lack et al., 2014). In particular, inorganic carbonates may generate spurious signals, soot may partly vaporize at the OC stage, materials such as tar balls or highly oxidized organics may generate EC signals, and inorganic compounds may catalyze the formation of EC or confound the optical quantification of pyrolysis (Corbin et al., 2020). It is also well known that different temperature ramp protocols lead to differences in the OC / EC ratio reported by TOA (e.g., Bautista et al., 2015; Schauer et al., 2003; Cavalli et al., 2010; Brown et al., 2017; Cheng et al., 2010; Giannoni et al., 2016; Wu et al., 2016; Cheng et al., 2012).

Less well studied are the uncertainties in TOA across multiple laboratories. Interlaboratory studies allow for an assessment of measurement reproducibility (changing laboratories, instruments, and operators), rather than simply repeatability (e.g., replicate measurements performed by the same operator). Here, the few reproducibility studies that exist have often focused on atmospheric aerosols or surrogates thereof. Schmid et al. (2001) analyzed urban air pollution samples from Berlin, Germany, using 9 different protocols obtained from 17 different laboratories. They reported relative standard deviations for between-laboratory uncertainty on their TC measurements of 6.7 %–11 %, with between-laboratory contributions making up 87 %–96 % of the overall variance. Schauer et al. (2003) evaluated EC and OC reproducibility for filter samples of Asian and North American air pollution, as well as secondary organic aerosol, reporting between-laboratory standard deviations of 12 %–22 % for EC and 3.6 %–12 % for OC. They additionally evaluated the reproducibility of the EC / OC division (split point) for various other samples, focusing on this ratio after identifying it as a major source of uncertainty. Ten Brink et al. (2004) sampled rural air pollution in Germany and analyzed the filters in four different laboratories, reporting less than 10 % variability in TC and EC. In a pan-European study, Panteliadis et al. (2015) gathered results from 17 different laboratories

to determine a reproducibility standard error of 20 %–26 % for EC and 12 %–15 % for TC. Finally, Brown et al. (2017) reported a combined standard error of < 13 % for a reproducibility study between four laboratories. The known technical shortcomings of TOA instruments cannot explain the magnitude of these uncertainties (Boparai et al., 2008).

We note that neither Schmid et al. (2001), Ten Brink et al. (2004), nor Panteliadis et al. (2015) presented a detailed statistical analysis of OC concentrations, and they reported up to a factor of 2 difference between OC measured by different protocols. This is related to the fact that the accurate quantification of OC in atmospheric samples is extremely difficult, due to the potential vaporization and/or adsorption of volatile organic compounds during and after sampling. This is especially true for low filter loadings, even when attempting to measure these artifacts (discussed below). This difficulty is one of the reasons that emissions testing protocols typically specify the removal of volatile OC by devices such as catalytic strippers, which remove all volatiles (typically at 350 °C) prior to filter collection. Consequently, any carbon measured as OC on downstream filters must either represent pyrolysis products or contamination. Importantly, Corbin et al. (2020) showed that once gas-phase contamination is accounted for, the remaining OC is also measured by in situ (filter-free) techniques and is, therefore, not a sampling or TOA artifact.

Overall, despite a very significant body of work on the fundamentals and statistical uncertainties behind TOA measurements, there have been few studies in which the sample was (i) non-volatile, (ii) taken from the same or identical filter, and (iii) of known composition. Here, we present an intercomparison study in which the same filters were punched six times for analysis by five different laboratories, after sampling aircraft engine exhaust treated at 350 °C with a catalytic stripper. Identical instrument models and protocols were used by all laboratories. Our study provides a general estimate of the between-laboratory uncertainty of TOA analyses from similar emissions tests and acts as a lower limit for the TOA reproducibility in atmospheric studies in which additional uncertainties are introduced (e.g., by way of having multiple sources or differing thermal protocols).

## 2 Methods

### 2.1 Experimental protocol

Sampling was performed in accordance with SAE ARP6320A (SAE, 2018), with the experimental setup shown schematically in Fig. 1. Emissions were collected from the exhaust of a helicopter turboshaft engine using a single-point sample probe, in a subsequent study to MANTRA (reported by Olfert et al., 2017), on the same model of engine and at the same facility. The sample stream was mixed with heated dilution air before passing through

a catalytic stripper (Model CS15, Catalytic Instruments). The sample flow was split to pass through a pair of Dekati diluters (DI-1000, operated with HEPA-filtered compressed air, where HEPA denotes high-efficiency particulate air) and

5 a pair of cyclones, each with a 1.0 μm cutoff at 50 L min$^{-1}$, before being directed through a sampling manifold. Samples were distributed from the manifold to a suite of instruments, including other instruments for online mass quantification (e.g., as in Corbin et al., 2020) and TOA. Particles for

TOA were collected on quartz filters in stainless-steel filter holders. The quartz filters were then sealed in Analyslide Petri dishes (catalog no. 28145-473) and kept at room temperature until analysis.

Samples were composed of 20 filters, with 5 respective

filters at mass concentrations of approximately 50, 100, 250, and 500 μg m$^{-3}$ (based on measurements made by an AVL Micro Soot Sensor on a separate parallel line connected to the sampling manifold). To compensate for the different mass concentrations used for loading the filters, the sampling time

durations were adjusted such that the mass loadings of nvPM on each filter were similar for all 20 filters. All nvPM samples were collected under high-power conditions for the Gnome engine. All of the samples loaded at nvPM mass concentrations of 50, 100, and 250 μg m$^{-3}$ were obtained with the en-

gine operating at a steady 22 000 rpm. To produce the higher nvPM concentration required for the samples loaded at a mass concentration of 500 μg m$^{-3}$, these samples were obtained with the engine operating at a steady 23 000 rpm. Saffaripour et al. (2020) demonstrated that there is no significant

change in the morphology of the particles from the same engine model for such modest changes in the rotation speed. Five laboratories compliant with ISO 17025 (demonstrating competence) for TOA were selected for this intercomparison. Each of the laboratories was instructed to take one (or two,

in the case of one laboratory) punches from each of the 20 filters. Seven punches were possible on each filter with an allowance of one spare punch per filter in addition to one (or two) per laboratory, arranged in a ring of six with one central punch, as shown in the inset to Fig. 1. Punch positions

on each sample were implicitly randomized by not otherwise providing further instruction to the laboratories. While this introduces a slight risk in the case of uneven filter loading, symmetry in the sampler and random filter orientations would minimize such risks in all but the center punch. The

loading of the filters was visually homogeneous, which further supports this decision. Further, even if there was a bias, for instance, due to handling of the filter, it is important to capture this as part of the interlaboratory variability, as this would be representative of real-world measurements. Quartz

filters adsorb gas-phase organic artifacts, and following the procedure outlined in Corbin et al. (2020), the data from TOA of the quartz filter in the second filter holder shown in Fig. 1 are used to correct the OC and TC measurements from the front filter for the gas-phase organics that were adsorbed on

the front filter.

The determination of TC, OC, and EC was quantified using the same TOA protocol, instrument model, and software settings for all participating laboratories. In all cases, analysis took place on Sunset Laboratory Model 5L analyzers (analogous interlaboratory comparisons have yet to be per- 60 formed on the other commercially available instrument, the Sunset Laboratory Model-4 Semi-Continuous OC-EC Field Analyzer CE3 ). The protocol for aircraft engine emissions, a refinement based upon NIOSH 5040 (SAE, 2018; Lobo et al., 2015b) with the final oxidizing temperature step at a 65 higher temperature of 930 °C and a longer duration to ensure complete oxidation of the particles, was used to perform the analysis, with the EC / OC split determined automatically by the instrument software (Calc405, Sunset Laboratory). The protocol and sample data are shown in Fig. 2. 70

Of the six sets of measurements considered, two belonged to a single laboratory and analyst; these are denoted in subsequent figures and discussion as Laboratory 1A and 1B. The remaining laboratories contributed a single set of data and are numbered in ascending order in terms of the average EC 75 measurement across all the filters. (Two of the laboratories were commercial service providers and did not contribute scientifically to the work.)

## 2.2 Statistical treatment

Results are analyzed using a hierarchical random effects 80 model. In this framework, measurements, $Y_{ijk}$, are modeled as a combination of effects:

$$Y_{ijk} = F_j + L_{ij} + E_{ijk}, \tag{1}$$

where $i$, $j$, and $k$ denote the $i$th laboratory, $j$th filter, and $k$th repeat. The remaining quantities are random variables 85 accounting for various effects or biases: the quantity $F_j$ is a filter-specific effect, accounting for any inconsistency in the loading of the filters; the quantity $L_{ij}$ is the effect or bias for each laboratory and represents a systematic shift in the measurements made by that laboratory for the $j$th filter; the 90 remaining term, $E_{ijk}$, represents the additional random error in the individual measurements reported by each laboratory, i.e., the mismatch between the expected laboratory bias and the actual measurement. As is a typical convention, uppercase letters are used here to denote a random variable, while 95 lowercase letters are used to denote the realization of the variable. Thus, $l_{ij}$ is the realization of $L_{ij}$ and corresponds to the bias specifically for the $i$th laboratory on the $j$th filter. This quantity will be a positive value if a laboratory has a bias above the filter-specific effect and vice versa. If there is 100 no such bias in any of the laboratories, all of the $l_{ij}$ values will be zero. The statistical model is shown schematically in Fig. 3 for a single filter.

A given laboratory reported their measurements, denoted as $y_{ijk}$ (i.e., a realization of $Y_{ijk}$), and an associated 105 laboratory-reported standard deviation, denoted as $\tilde{s}_{ij}$. The uncertainty values reported by the different laboratories are

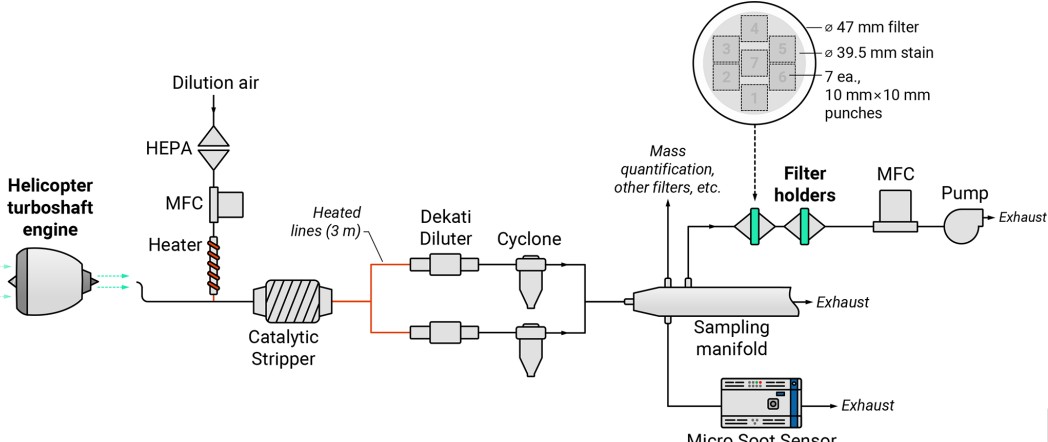

**Figure 1.** Schematic of the experimental setup in which emissions from a helicopter turboshaft engine transit to filter holders containing the quartz filters used for thermal–optical analysis (TOA). Cyclones had a $1.0\,\mu m$ cutoff at $50\,L\,min^{-1}$, with the actual sample flow rate for each cyclone being $56\,L\,min^{-1}$. MFC stands for mass flow controller, while HEPA refers to a high-efficiency particulate air filter. The diluter–cyclone system is consistent with SAE (2018). The inset in the top right depicts the punch positions on the filter. Note that the angular position of the punches on the filter was not constrained.

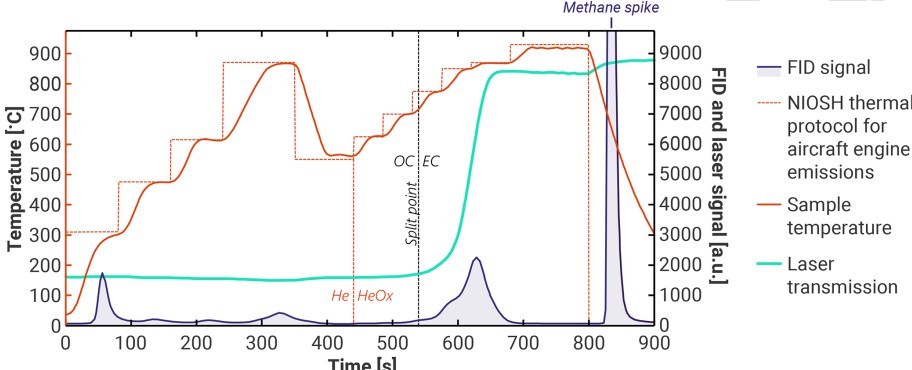

**Figure 2.** Representative example of a TOA thermogram for nvPM emissions collected from the engine used in this study. Shown are the thermal protocol for aircraft engine emissions (SAE, 2018; Lobo et al., 2015b), the sample temperature, the flame ionization detector (FID) signal, and the laser transmission measurement. HeO$_x$ corresponds to 2 % oxygen in He. FID is a flame ionization detector. The methane spike corresponds to the introduction of methane that is used for calibration after analysis.

automatically generated by the analysis software on the Sunset Laboratory Model 5L, which is a combination of the limit of detection of the instrument ($0.2\,\mu g\,cm^{-2}$) and a percentage based upon prior statistical analysis of duplicate filter punches ($\pm 5\,\%$). For these reasons, the laboratory-reported uncertainties do not truly represent repeatability, incorporating a wider range of effects. Further, given limitations in the number of punches available for each filter in this study and the fact that the test is destructive, only a single measurement is available for each laboratory–filter combination here. As such, the model is hereafter stated without the subscript $k$ dependence:

$$Y_{ij} = F_j + L_{ij} + E_{ij}. \qquad (2)$$

The use of single measurements also complicates a simple interpretation using ISO 5725-2 (ISO, 2019), as was applied by Panteliadis et al. (2015).

Rather, a Bayesian method is employed. Realizations of the random effects models are obtained using a Markov chain Monte Carlo (MCMC) approach, similar to the method presented by Melanson et al. (2018) and in the direction of the method used by Conrad and Johnson (2019). MCMC seeks to find the range of inputs, in this case the magnitude of various effects and uncertainties, that would cause some distribution of the observed measurements. The approach in this work attempts to overcome the limitations in the preceding paragraph by noting that, while there was variability across the filters, the laboratory effects are assumed to be consistent across the different filters (i.e., the laboratory effects are

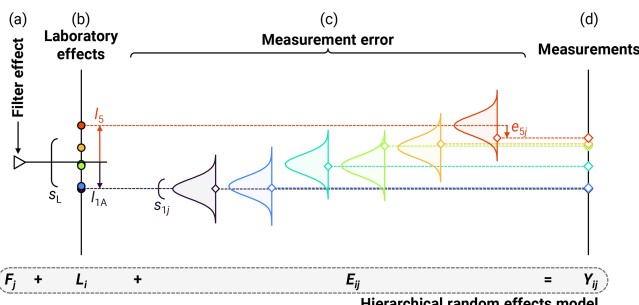

**Figure 3.** Schematic demonstrating the hierarchical random effects model used in the present analysis. Data correspond to a single filter computed using the Markov chain Monte Carlo (MCMC) procedure described in the text.

not specific to a given filter), such that the model is further simplified to

$$Y_{ij} = F_j + L_i + E_{ij}, \qquad (3)$$

removing the $j$th filter dependence from $L$. Combining this with an assumption of normally distributed measurement errors, a likelihood can then be stated as

$$Y_{ij} \sim \mathcal{N}\left(f_j + l_i, s_{ij}^2\right), \qquad (4)$$

where the $s_{ij}$ values are unknown within-laboratory uncertainties. This likelihood relates the various effects to the observed measurements. To restrict the solution space and improve convergence of the sampling algorithm, priors are also applied (encoding approximate information known before the statistical analysis) to these quantities, which are summarized in Table 1. Exponential priors are used for the variances and Gaussian priors for the effects. These correspond to maximum entropy priors for variables given that variances have a point estimate and effects generally have point estimates and a spread, a priori. The sets of $s_{ij}$, $f_j$, $l_i$, and $\tilde{s}_L$ are sampled as part of the MCMC procedure, where the latter quantity, $\tilde{s}_L$, used in the prior placed on $L_i$, is treated as a nuisance parameter (i.e., it is allowed to change and contribute to variability in the sampling but is not explicitly included in the reported output). All of the data are considered in a single MCMC run, rather than separating the data into levels as in ISO 5725-2. To minimize the impact of a large burn-in period for the MCMC, the set of $f_j$ was initiated about the average of $y_{ij}$ for a single filter, while $l_i$ was initiated about the average $y_{ij}$ over all of the filters after subtracting the average $f_j$. A total of 25 000 samples were generated, after thinning the MCMC data by a factor of 20 (to avoid short-range correlation in the samples) spread across four independent chains. MCMC samples were realized using the Just Another Gibbs Sampler (JAGS) code (Hornik et al., 2003). Visual inspection of the samples indicated that the chains had converged. Further increasing the number of samples did not have an impact on the statistical outcomes. A brief comparison of

the MCMC method with an application of the ISO 5725-2 method is presented in Sect. S1 the Supplement, with overall reproducibility holding similar values in most instances to those derived with the current method but with a different breakdown of the uncertainties. Note that the repeatability variance, $s_r^2$, is estimated from this procedure using the average of the within-laboratory variances, $s_{ij}^2$, roughly consistent with the ISO method, except that it is estimated in the MCMC procedure instead of computed directly from repeat measurements. Table 2 summarizes the different variances used in this work.

## 3 Results

### 3.1 Statistical analysis of EC, OC, and TC

Figure 4 shows a sample of the results for EC for 5 of the 20 filters, corresponding to the $100\,\mu\mathrm{g\,m^{-3}}$ mass concentration case. In Fig. 4, laboratories are ordered according to the median EC measured over all 20 filters. This order is roughly respected across all of the filters. Results are generally consistent with the remaining 15 filters (not shown), although filter-to-filter differences exceeded the range shown in this subset of the filters in some cases. Uncertainties did not exhibit a trend with mass concentration or the measured value for EC, OC, or TC. Nominal or consensus values were determined by taking the mean of the filter effects as determined by the MCMC procedure.

Uncertainties, alongside their decomposition into their respective components, are reported in Table 3. The uncertainties CE4 in the subsequent discussion are expressed as coefficients of variation (or relative standard errors), where the coefficient of variation is the square root of the corresponding variance divided by the nominal value of EC, OC, and TC, as appropriate. Further, *expanded* uncertainties, defined as an interval about the result of a measurement that may be expected to encompass a large fraction of the distribution of values that could reasonably be attributed to the measurand (JCGM, 2008), are used. The expanded uncertainties are defined by a coverage factor, $k$, defined as a numerical factor used as a multiplier of the combined standard uncertainty in order to obtain an expanded uncertainty. Following convention, $k = 2$ is used for the expanded uncertainty throughout this work, which is roughly equivalent to a 95 % confidence interval.

Figure 5 shows a decomposition of the two kinds of within-filter uncertainties (within- and between-laboratory uncertainties), averaged over all of the filters and laboratories and presented as a proportion of the observed variance. For EC and TC, uncertainties are dominated by between-laboratory contributions, making up roughly 85 % of the observed variance in both cases. This is a clear indication that repeatability is a poor measure of the overall uncertainties in these measurements and that there is indeed a requirement

**Table 1.** Quantities related to the statistical treatment, including those on which likelihood and priors are directly stated. Note that measurements, $y_{ij}$, are the input to the MCMC procedure and are, thus, not sampled. The likelihood corresponds to the assumed form for the distribution of a given quantity, which is the distribution to be reproduced by the MCMC sampling procedure.

| Quantities computed or sampled | Likelihood (assumed form) | Priors |
|---|---|---|
| $l_i, f_j, \tilde{s}_L, s_{ij}$ | $Y_{ij} \sim \mathcal{N}\left(f_j + l_i, s_{ij}^2\right)$ | $L_i \sim \mathcal{N}\left(0, \tilde{s}_L^2\right)$ |
| | | $F_j \sim \mathcal{N}\left(m_j, s_{F,j}^2\right)$ |
| | | $\tilde{S}_L \sim \mathrm{Exp}\left(1/\mathrm{MAD}(y_{ij})\right)^*$ |
| | | $S \sim \mathrm{Exp}\left(1/\tilde{s}_{ij}\right)^*$ |

* $\mathrm{Exp}(\lambda)$ corresponds to an exponential distribution with a rate parameter $\lambda$ and is the maximum entropy prior for random variables that only have a point estimate. MAD denotes the median absolute deviation.

**Table 2.** Variances used in the work, their computation, and their corresponding symbol.

| Uncertainty component | Estimation procedure | Symbol |
|---|---|---|
| Within-laboratory uncertainty | MCMC, direct sampling | $s_{ij}$ |
| Laboratory-reported uncertainty | Reported by laboratories (average of variance)* | $\tilde{s}_{ij}, (\tilde{s})^*$ |
| Repeatability uncertainty (approx.) | Average $s_{ij}^2$ across laboratories | $s_r$ |
| Between-laboratory uncertainty | MCMC, standard deviation of $l_i$ | $s_L$ |
| Prior for between-laboratory uncertainty | MCMC, direct sampling | $\tilde{s}_L$ |
| Reproducibility uncertainty | $s_R^2 = s_L^2 + s_r^2$ | $s_R$ |
| Reproducibility uncertainty for a single laboratory | $s_{R,ij}^2 = s_L^2 + s_{ij}^2$ | $s_{R,ij}$ |
| Between-filter uncertainty | MCMC, standard deviation of $f_j$ | $s_F$ |
| Total uncertainty | $s_{TOT}^2 = s_F^2 + s_R^2$ | $s_{TOT}$ |

* The value in parentheses is the average laboratory-reported uncertainty, computed by averaging the variance from each laboratory and filter.

for larger uncertainties to account for true variability in the TOA method.

Figure 6 complements Fig. 5 with a plot of the measurements for each laboratory across the filters and normalized by the filter effects, $f_j$. Filters are sorted in ascending order of the nominal value for EC, OC, and TC, respectively, such that the filter order differs between the panels. Trends in Fig. 6 for EC and TC were similar, given that EC concentrations were typically double the OC concentrations. We note that the uncertainty in TC is smaller than the uncertainty in either OC or EC, as OC and EC are calculated by splitting TC into two parts. Additional uncertainty arises due to this split, which is determined from the laser transmission through the filter, a software algorithm, and the estimation of how much OC had charred to form EC. Results for EC and TC exhibit a consistent bias (or systematic error; JCGM, 2008) across the different filters; that is, a laboratory that reported a single above-average value generally did so for all of its reported values. For example, Laboratory 5 produced EC and TC values consistently above the other laboratories, whereas Laboratory 1 (in both the 1A and 1B samples) produced EC and TC values consistently below the other laboratories. Consistency within each laboratory drives smaller within-laboratory contributions (as each laboratory was consistent with itself), with a corresponding expansion of the between-laboratory contributions to account for the remaining spread. The observed biases in the data may also give insight into the physical causes of these uncertainties. For example, minor biases in calibration would lead to the observed systematic errors, while random operator error would not. Other potential sources of error (e.g., in terms of FID response) have been discussed in detail elsewhere (Boparai et al., 2008).

Laboratory-reported uncertainties always exceeded the repeatability computed by the MCMC procedure. The reason for this becomes apparent from the data. Laboratory-reported uncertainties, also shown in Fig. 5, appear to accommodate all of the within-laboratory contributions as well as some of the between-laboratory contributions. We denote the discrepancy between the reproducibility and the laboratory-reported uncertainties as *dark* (Thompson and Ellison, 2011) contributions, given that such contributions would be hidden outside of an interlaboratory study and so as to distinguish them from the more precise between-laboratory contributions determined by the MCMC procedure. While the laboratory-reported variances are just slightly below the combined MCMC between-laboratory variance and within-laboratory variance for TC and for OC, the laboratory-reported variance

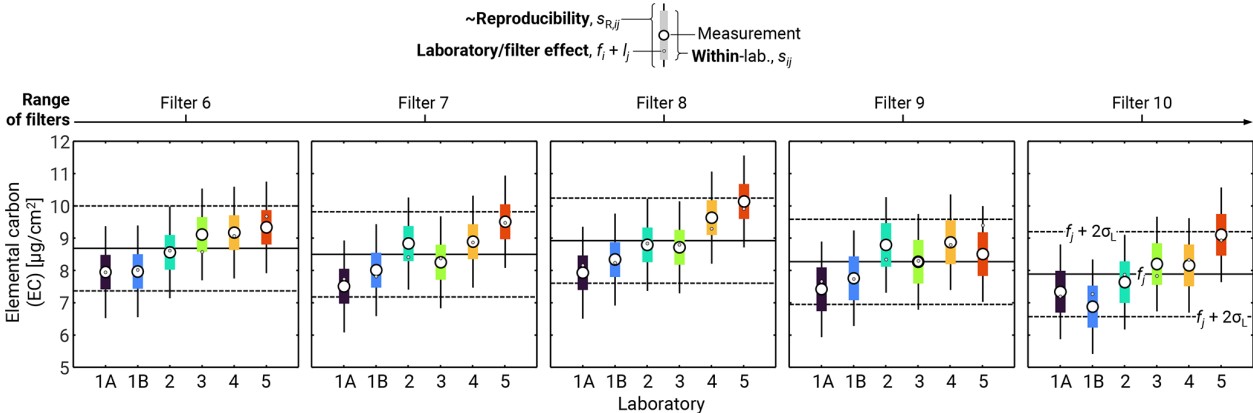

**Figure 4.** Sample results for the MCMC uncertainty procedure. Open circles correspond to laboratory-reported data, while small, filled circles correspond to the combined filter and laboratory effects. Bars correspond to within-laboratory uncertainties, $s_{ij}$, while whiskers correspond to reproducibility, including the average within- and between-laboratory uncertainties combined in quadrature as $s_{\mathrm{R},ij} = \left(s_{\mathrm{r},ij}^2 + s_{\mathrm{L}}^2\right)^{1/2}$. Horizontal, solid lines correspond to the realized filter effect, while dashed black lines correspond to expanded reproducibility intervals ($k = 2$). Results shown are for elemental carbon and the $100\,\mu\mathrm{g\,m^{-3}}$ case. Vertical axes are identical across all of the panels.

**Table 3.** Breakdown of uncertainties in the TOA measurements, stated as expanded coefficients of variation (or relative standard errors, i.e., the expanded standard error divided by the nominal value of EC, OC, and TC) for a coverage factor of $k = 2$. Reproducibility variance, $s_{\mathrm{R}}^2$, is a combination of the within- and between-laboratory variances (i.e., $s_{\mathrm{R}}^2 = s_{ij}^2 + s_{\mathrm{L}}^2$). The total row corresponds to a combination of reproducibility and between-filter uncertainties, again by summing the variances.

| Uncertainty component | Symbol | Expanded coefficient of variation ($k = 2$) [%]* | | | | |
|---|---|---|---|---|---|---|
| | | EC | OC | TC | EC / OC | EC / TC |
| Within-laboratory (repeatability) uncertainty | $s$ | 6.8 | 12.5 | 4.7 | 3.8 | 2.7 |
| Between-laboratory uncertainty | $s_{\mathrm{L}}$ | 16.0 | 7.3 | 11.7 | 17.8 | 6.6 |
| Reproducibility uncertainty | $s_{\mathrm{R}}$ | 16.5 | 8.0 | 12.1 | 18.6 | 6.8 |
| Between-filter uncertainty | $s_{\mathrm{F}}$ | 21.1 | 8.6 | 14.8 | 23.1 | 9.1 |
| Total uncertainty | $s_{\mathrm{TOT}}$ | 26.8 | 11.8 | 19.1 | 29.7 | 11.3 |
| Laboratory-reported uncertainty | $\tilde{s}$ | 12.4 | 14.3 | 13.1 | – | – |

* Coefficients of variation are stated using the nominal EC, OC, and TC measurement values of 8.1, 4.7, and 12.9 $\mu\mathrm{g\,cm^{-2}}$ and EC / OC and EC / TC ratios of 1.74 and 0.63.

grossly underestimates the combined MCMC variances for EC, requiring a further 93 % of the laboratory-reported variance to match the MCMC combined variances.

Unlike EC and TC, Fig. 5 indicates that OC uncertainties are driven by within-laboratory contributions. This is reflected in the fact that consistent biases were less evident for OC in Fig. 6. In other words, repeatability within a laboratory is of the same order of magnitude as the overall reproducibility for OC. Again, laboratory-reported uncertainties seem to account for the overall reproducibility in OC – in this case, even accommodating the between-filter variability.

For the respective EC, OC, and TC measurements, the reproducibility reported here is smaller than that reported by Panteliadis et al. (2015), who provided values equivalent to expanded ($k = 2$) uncertainties of 40 %–50 % and 24 %–30 % for EC and TC, respectively, using the ISO 5725-2 method

(ISO, 2019). This is most likely due to greater variability between the atmospheric samples measured by Panteliadis et al. (2015), relative to the single aerosol source and volatile removal device in our study. Our within-laboratory relative standard errors are also smaller than those measured by a single laboratory in Conrad and Johnson (2019). Those authors provided expanded uncertainties of 20 %, 44 %, and 17 % for EC, OC, and TC (from Table 2 in that work) relative to the 6.8 %, 13 %, and 4.7 % observed in the present work. Conrad and Johnson (2019) also determined that TC is the most repeatable, whereas OC is the least repeatable, again consistent with the current observations. The relative breakdown of within- and between-laboratory contributions to the uncertainties for TC here are also similar to the relative contributions observed by Schmid et al., 2001, although the uncertainties here are again smaller (expanded between-

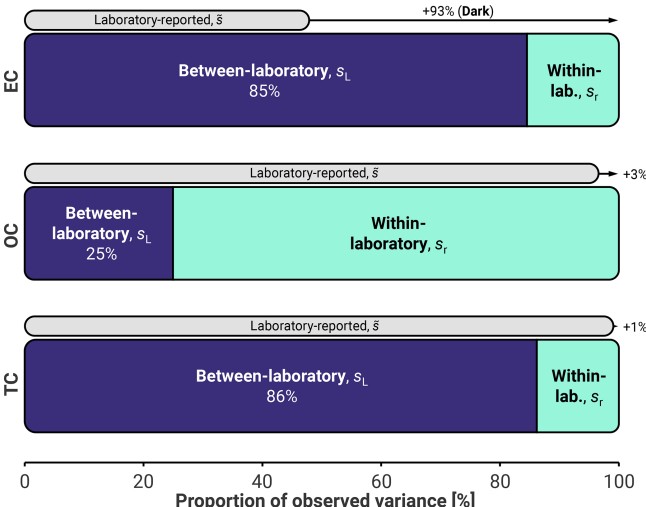

**Figure 5.** Breakdown of the within-filter variance in the TOA measurements into the between-laboratory variance, $s_L^2$, and within-laboratory variance, $s_r^2$, stated as a proportion of the overall within-filter variance, such that all span 0 %–100 %. Also shown is the corresponding average variance reported by the laboratory, $\bar{s}^2$. The reader is referred to Table 3 for the numerical values of the uncertainties. Percentages in dark bars correspond to the required increase in uncertainties over the within-laboratory values. Note that the difference between the overall bar width and the laboratory-reported uncertainties was computed on a per-filter basis (rather than using the values in Table 3) and was not allowed to be negative. For this reason, laboratory-reported bars in this figure are always smaller than 100 %. CE5

laboratory uncertainties of 18 % in Schmid et al., 2001, versus 12 % in this work). These collective observations indicate that the current measurements share many of the same trends as previous works, but uncertainties in this work are consistently smaller. We hypothesize that our smaller uncertainties are primarily due to the removal of volatile organics with a catalytic stripper, as organics are subject to transformation and mass loss during handling and storage. As we observed lower uncertainties for TC than other studies, our lower uncertainties for EC and OC cannot be attributed to the EC–OC split alone. This is further supported by the lack of negative correlation between EC and OC (see Sect. 3.2), indicating that the split point was determined reliably. Further, it is likely that the use of a single particle source, a single thermal protocol, a single instrument model, and a common version of software all contribute to the smaller uncertainties observed in this study.

Overall, for our samples, expanded ($k = 2$) uncertainties for reproducibility in EC, OC, and TC for a given filter are 17 %, 8.0 %, and 12 % of the nominal values, respectively.

## 3.2 Statistical analysis of the EC / OC and EC / TC ratios

Little to no correlation, $R$, was observed between EC and OC measured by the different laboratories ($R_{\text{ECOC}} = 0.11$), while TC was dominated by, and thus highly correlated with, the EC contributions ($R_{\text{ECTC}} = 0.94$). Combining this with the fact that the measured EC showed consistent biases across the laboratories, it is logical that this is equally reflected in the TC results. OC and TC were poorly correlated ($R_{\text{OCTC}} = 0.43$), given that TC incorporates but is not dominated by OC. The low level of correlation between EC and OC indicates that the split point is unlikely to be the leading driver of variabilities in the results, as this would result in a negative correlation between EC and OC, where more of the total carbon is attributed to one of the components at the cost of the other.

Unlike the absolute values for EC, OC, and TC, the EC / OC ratio is expected to be similar across all of the filters, regardless of loading, and is a widely used quantity for characterizing the particles emitted. For the EC / OC ratio, simple propagation of errors yields the following (Sipkens et al., 2023; JCGM, 2008):

$$\text{var}(\text{EC} / \text{OC})$$
$$= \left(\frac{\text{EC}}{\text{OC}}\right)^2 \left[ \frac{1}{(\text{EC})^2} \text{var}(\text{EC}) + \frac{1}{(\text{OC})^2} \text{var}(\text{OC}) - \frac{2}{(\text{EC})(\text{OC})} \text{cov}(\text{EC}, \text{OC}) \right]. \tag{5}$$

As noted above, EC and OC are not significantly correlated for these measurements, such that the covariance term can be neglected. Overall, the EC / OC ratio is $1.74 \pm 0.52$ ($k = 2$) for the full set of measurements. The expanded uncertainties for the EC / OC ratio, including the contributions from the different sources of variability, are also reported in Table 3. Expanded uncertainties are in terms of the coefficient of variation (or relative standard error), with a coverage factor of $k = 2$. This produces a relatively uniform estimate of the expanded uncertainty for reproducibility in the EC / OC ratio across all of the measurements, at 19 % of the nominal value. This is comparable to the laboratory-reported uncertainties. Between-filter variability was significant, increasing the expanded uncertainties to 30 % of the nominal value. Note that this is larger than the uncertainties in the individual EC and OC measurements, as it incorporates uncertainties in both EC and OC measurements, as it incorporates uncertainties in both EC and OC at the same time.

In these data, Laboratory 5 produced an EC / OC value consistently above the other laboratories, a consequence of measuring higher-than-average EC in combination with a slightly lower-than-average OC. There was also some trend in EC / OC with mass concentration and sampling time, for all laboratories, indicated by the generally increasing filter number on the $x$ axis of Fig. 7. This results from a similar slight increase in EC and a slight decrease in OC as the sam-

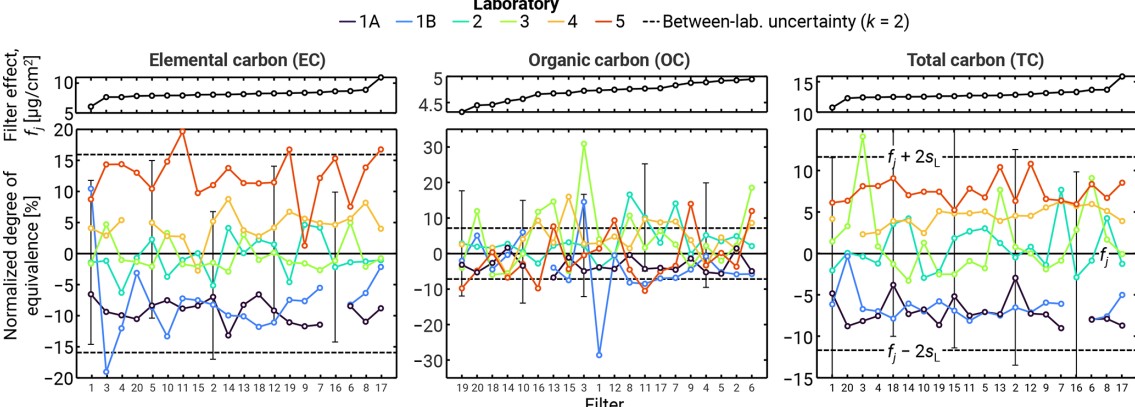

**Figure 6.** Laboratory measurements across the different filters, showing the associated effects (i.e., biases) in the data. In each case, filters are re-sorted such that the consensus values for the filters are monotonically increasing for each of the measurands. As such, the order of filters is not the same across the panels. Upper panels show the consensus values for the different filters. Bottom panels show measurements from each laboratory normalized by those consensus values. Breaks in lines correspond to results that were not available. Error bars in the lower panels correspond to expanded ($k = 2$) uncertainties reported by the laboratories and, while only included for select points, were similar in magnitude across all of the data.

pling period decreases. As this effect was minor, our preceding discussion summarized the data using the means of EC, OC, TC, and the EC / OC ratio.

Similar principles can be applied to the EC / TC ratio, where

$$\text{var}(\text{EC}/\text{TC})$$
$$= \left(\frac{\text{EC}}{\text{TC}}\right)^2 \left[ \frac{1}{(\text{EC})^2} \text{var}(\text{EC}) + \frac{1}{(\text{TC})^2} \text{var}(\text{TC}) \right.$$
$$\left. - \frac{2}{(\text{EC})(\text{TC})} \text{cov}(\text{EC}, \text{TC}) \right]. \tag{6}$$

This time, the covariance term will necessarily be significant, as TC is largely composed of EC. The present analysis uses a correlation of $R_{\text{ECTC}} = 0.94$, as noted above, and rephrases Eq. (6) in terms of the correlation:

$$\text{var}(\text{EC}/\text{TC})$$
$$= \left(\frac{\text{EC}}{\text{TC}}\right)^2 \left[ \frac{1}{(\text{EC})^2} \text{var}(\text{EC}) + \frac{1}{(\text{TC})^2} \text{var}(\text{TC}) \right.$$
$$\left. - \frac{2 R_{\text{ECTC}} [\text{var}(\text{EC}) \text{var}(\text{TC})]^{1/2}}{(\text{EC}), (\text{TC})} \right]. \tag{7}$$

The resultant uncertainties are quite small, due to the high degree of correlation, amounting to expanded ($k = 2$) uncertainties of 6.8 % within a given filter and 11 % when adding between-filter variability. A majority of this variability stems from between-laboratory variability, consistent with the observations for EC and TC.

## 4 Conclusions

This work investigated the between-laboratory uncertainties associated with thermal–optical analysis (TOA) applied to aircraft engine particulate emissions. These conditions represent optimal samples for TOA, in that they are primarily composed of combustion particles that are stripped of their volatile components. Uncertainties are not expected to be related to the split point, due to a lack of correlation between EC and OC (where a reduction in OC results in an increase in EC).

EC and TC measurements are highly correlated with the laboratory (i.e., reflected by a fixed bias or systematic error), with some laboratories measuring consistently above or below the average. These laboratory biases suggest a potential link to laboratory-specific calibration that affects the EC (and, by extension, the TC) measurement. This results in EC and TC uncertainties being dominated by between-laboratory contributions ($\sim 85$ % of the variance). Further, replicates, i.e., repeat measurements by a single laboratory, are unlikely to properly capture these uncertainties. For data sets comparable to ours (i.e., PM dominated by soot, treated to remove volatile organic carbon, and containing negligible elemental impurities), net expanded ($k = 2$) relative standard errors of 17 % for EC, 8.0 % for OC, 12 % for TC, 19 % for the EC / OC ratio, and 6.8 % for the EC / TC ratio are expected and account for reproducibility. These values correspond to a lower limit on the uncertainties for EC, OC, and TC, given the use of a single particle source, a single thermal protocol, a single instrument model, and a catalytic stripper to remove volatile organics. This expanded uncertainty should be used in future measurements with this test method. For application to the calibration of instruments to measure

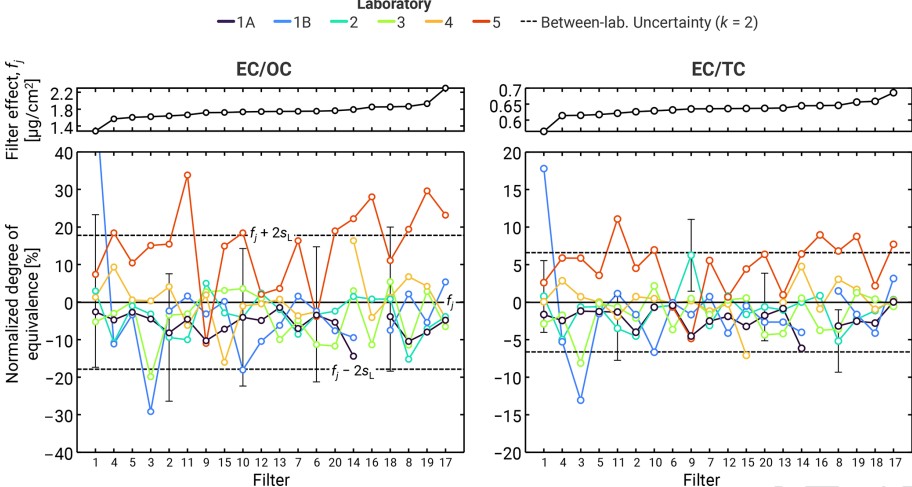

**Figure 7.** Variability in the EC / OC and EC / TC ratios over the measurements. Uncertainty intervals correspond to the between-laboratory expanded uncertainties at a level of $k = 2$. Upper panels show the consensus EC / OC values against which the filters are sorted. Bottom panels show the measurements from each laboratory normalized by those consensus values. As for Fig. 6, error bars in the lower panels correspond to expanded ($k = 2$) uncertainties reported by the laboratories and, while only included for select points, were similar across all of the data.

the mass concentration of nvPM emissions from aircraft engines, the expanded uncertainties for EC (17 %) and for the EC / TC ratio (6.8 %) are the most significant quantities.

The authors see limited scope for reducing the uncertainties in TOA through refinements to the calibration procedures and quality controls. While promising alternatives to TOA are emerging for calibration of instruments, such as the centrifugal particle mass analyzer electrometer reference mass standard (CERMS) (Titosky et al., 2019; Corbin et al., 2020), the corresponding interlaboratory variability in these alternatives has yet to be validated and should be a topic of future work.

The treatment in this work does not directly address the interpretation of OC and EC concentrations reported by TOA, nor does this work evaluate the accuracy of the TOA TC concentration (e.g., by indicating traceability to an International System of Units unit). Rather, this work addresses metrological reproducibility of the TOA method by comparing results from the same sample, measured by different laboratories and analysts.

*Data availability.* A simplified form of the raw data, including the laboratory-reported measurements and uncertainties, has been included in the Supplement as CSV files. One file is provided for EC, OC, and TC measurements, respectively. The first columns contain information about the laboratory and whether or not the row corresponds to a measurement ("$y$") or laboratory-reported uncertainty ("std"). Each column contains the results for a different filter.

*Supplement.* The supplement related to this article is available online at: https://doi.org/10.5194/amt-17-1-2024-supplement.

*Author contributions.* TAS: data curation, formal analysis, methodology, software, visualization, and writing – original draft, review, and editing. JCC, BS, SG, BTB, AF, and MJ: investigation and writing – review and editing. PL: supervision, project administration, and writing – review and editing. GJS: conceptualization, investigation, methodology, project administration, supervision, and writing – review and editing.

*Competing interests.* The contact author has declared that none of the authors has any competing interests.

ther geographical representation in this paper. While Copernicus Publications makes every effort to include appropriate place names, the final responsibility lies with the authors.

*Acknowledgements.* We are grateful to the Rolls Royce team for their support with respect to obtaining these samples. This work was supported by the Transport Canada, Environmental Protection and Standards, Civil Aviation program. Filter analysis was supported in part by the Swiss Federal Office of Civil Aviation's "EMPAIREX" project. We also thank Jason Olfert for useful discussions on the statistical modeling presented in this work.

*Financial support.* This research has been supported by Transport Canada (grant no. A1-023090) and the Bundesamt für Zivilluftfahrt (grant no. SFLV-2015-113).

*Review statement.* This paper was edited by Pierre Herckes and reviewed by two anonymous referees.

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

**Remarks from the language copy-editor**

**Remarks from the typesetter**