# Peer review of "Quantifying the uncertainties in thermal-optical analysis of carbonaceous aircraft engine emissions: An interlaboratory study"

_Atmospheric Measurement Techniques, 2024_

## Author Comment (AC1)

This study comprehensively evaluates the previously poorly quantified inter-laboratory uncertainty in TOA analysis used to derive EC, OC, and TC. Aerosol exhaust from a helicopter engine was collected on 20 filters in accordance with regulatory civil aviation specification for nvPM mass instrument calibration. The comprehensive analysis presented in this study underscore the importance of including the inter-laboratory contribution to EC, OC, and TC uncertainties as it was found to be significant. I enjoyed reading your manuscript, which is well written, includes informative figures, and provides a detailed description of the methods and uncertainty analysis. However, I felt that the implications and limitations of your work were not discussed extensively. Below are a few general comments and minor suggestions for the authors to consider.

**Minor comments:**

- Line 19: You've omitted to say that regulatory aircraft nvPM mass can be directly calibrated on a gas turbine, not just a diffusion flame.

  *In the regulatory requirements, the calibration must be performed with a Diffusion Flame Combustion Aerosol Source (DFCAS), which can include gas turbine sources, as the reviewer notes. The text has been revised accordingly:*

  *"The real-time instruments used for regulatory measurements of aircraft engine non-volatile particulate matter (nvPM) mass emissions are required to be calibrated to the mass of EC determined by TOA of the filter-sampled emissions of a diffusion flame combustion aerosol source (DFCAS)."*

- Line 27: "*Uncertainties were a little larger for EC than for OC*"; I suggest merging with previous sentence and directly report OC uncertainty rather than using "*a little larger*".

  *Following changes to the statistical model, this text no longer appears in the mansucript.*

- Line 200-204: It is unclear to me how the 'dark inter-laboratory uncertainty', as shown in Figure 5 and Table 2, could be lower for TC than for EC, given that TC is derived from EC. Can you clarify and discussed in the main text if relevant?

  *In reality, the TOA method measures TC directly, and then assigns a split point in the analysis to separate the EC from the OC. Thus, while TC may appear to be the sum of OC and EC, there will be greater certainty with TC, as it is directly measured, and less certainty with OC and EC, as these quantities are assigned values based on the laser transmission through the filter, the software algorithm, and the estimation of how much OC had charred to form EC. Uncertainty in this estimation can lead to a bias in OC and EC, as*

*whichever direction the split point between the two moves, one will increase and the other will decrease. We have added a clarifying note to the manuscript to this effect:*

*"It is worth nothing that, while TC may appear to be the sum of OC and EC, there will be greater certainty with TC, as it is a directly measured quantity, and less certainty with OC and EC, as these quantities are assigned values based on the laser transmission through the filter, the software algorithm, and the estimation of how much OC had charred to form EC."*

*It is also worth noting that the previous statistical model made use of the laboratory-reported uncertainties, with the inter-laboraotry uncertainties simply making up the balance. Thus, if the laboratory-reported uncertainties are larger for TC, which they were, then the remainin proportion attributable to between-laboratory/dark uncertainties will also be smaller. The statistical model in the revised manuscript now attempts to directly infer the within-laboratory uncertainties, rather than rely on laboratory-reported uncertainties (which instead act as prior information). This is more intutive and was one of the reasons for the change to the statistical model. The result is that the proportions of the variance are now largely independent of the laboratory-reported uncertainties, which, in turn, results in proportions that are more-or-less similar between EC and TC, matching the reviewer's expectations (see revised Fig. 5, even if uncertainties in OC do, to some extent, propagate into TC as well).*

- The sections 3.1 and 3.2 titles aren't very clear, I suggest renaming them (for example 3.1 could be EC, OC and TC uncertainty and 3.2 could be EC/OC uncertainty).

  *The titles of these sections have been updated. Given that the sections present a broader analysis of the quantities, the sections have been renamed to "Statistical analysis of EC, OC, and TC" and "Statistical analysis of EC/OC and EC/TC ratios"*

- Line 287-288: *"uncertainties are poorly captured by existing estimates for these measurements"*; Can you clarify that you mean by existing estimates? Do you mean lab-reported or what has been reported in the literature or both? Can you also expand on *"except perhaps for measurements of OC"* because I didn't get that impression from your literature review in the introduction.

  *The associated statements referred to laboratory-reported uncertainties, rather than values from the literature. Thus, comments about OC do not reflect on previous studies.*

  *The updated statistical model better differentiates laboratory-reported uncertainties against within-laboratory and between-laboratory estimates. Much of the confusing language around this distinction has either been removed or entirely rewritten.*

*New statements related to this comment now read:*

*"Laboratory-reported uncertainties always exceeded the repeatability as computed by the MCMC procedure. The reason for this becomes apparent from the data. Laboratory-reported uncertainties, also shown in Figure 5, appear to accommodate all of the within-laboratory contributions as well as some of the between-laboratory contributions. We denote the discrepancy between the reproducibility and the laboratory-reported uncertainties as dark (Thompson and Ellison, 2011) contributions, given that such contributions would be hidden outside of an inter-laboratory study and so as to distinguish them from the more precise between-laboratory contributions determined by the MCMC procedure. While the laboratory-reported variances are just slightly below the combined MCMC between-laboratory variance and within-laboratory variance for TC and for OC, the laboratory-reported variance grossly underestimates the combined MCMC variances for EC, requiring a further 93% of the laboratory-reported variance to match the MCMC combined variances."*

*"Again, laboratory-reported uncertainties seem to account for the overall reproducibility in OC, in this case even accommodating the between-filter variability."*

- L304: It would be useful to introduce the term "metrological" earlier in the manuscript to ensure that all readers are familiar with the term and its significance.

  *Text has been added to the introduction to this effect:*

  *"However, questions remain open regarding the uncertainties and associated metrology (referring to the establishment of uncertainties by way of inter-laboratory comparisons and the traceability) of these measurements."*

**General comment:**

- Are there any instruments other than the Sunset 5L commercially available and used by competent laboratories to make TOA measurements? Could your reported uncertainties be specific to that instrument? I suggest discussing this where relevant.

  *To our knowledge, TOA measurements performed by competent laboratories are only made by Sunset Laboratory instruments including the manual Model 5L Lab OCEC Analyzer and the Model-4 Semi-Continuous OC-EC Field Analyzer. We are not aware of alternate commercially available analyzers that are compliant with the requirements of SAE ARP6320A or the regulatory requirements for aviation nvPM emissions.*

*The reported uncertainties are specific to the Sunset 5L. Similar ILCs remain to be performed for the Model-4 Semi-Continuous OC-EC Field Analyzer. This has been explicitly noted in the experimental protocol section:*

*"In all cases, analysis took place on Sunset Laboratory Model 5L analyzers (analogous ILCs have yet to be performed on the other commercially-available instrument, the Sunset Laboratory Model-4 Semi-Continuous OC-EC Field Analyzer)."*

- Something that caught my attention is that your calculated cumulated uncertainties (e.g., 26% (k = 2) for EC) from your highly controlled study (i.e., nvPM, identical filters, known composition) are higher than that reported from other studies (Schauer et al., Ten Brink et al, Panteliadis et al., Brown et al.), yet you mentioned that atmospheric samples from these studies should have higher uncertainties than yours. Why do you think you estimated a higher overall uncertainty than say, Panteliadis et al.? It would be interesting to discuss these differences in more details.

  *The uncertainties here are indeed lower than those of Panteliadis and co-workers. The percieved discrepency may be due to the use of k = 2 expanded uncertainties in this work, while those by Panteliadis and other authors often directly report standard errors without a coverage factor (equivalent to k=1). We have updated the manuscript to make the comparison to the previous literature more direct and to be more consistent in the use of coverage factors. Of particular relevance, we have added the following paragraph, which also now makes referencce to Schauer et al.:*

  *"For all three of EC, OC, and TC, reproducibility reported here is smaller than that reported by Panteliadis et al. (2015), who provided values equivalent to expanded (k = 2) uncertainties of 40–50 % and 24–30 % for EC and TC, respectively, using the ISO 5725-2 method (ISO, 2019). This is most likely due to a greater variability between the atmospheric samples measured by Panteliadis et al., relative to the single aerosol source and volatile removal device in our study. Our within-laboratory relative standard errors are also smaller than those measured by a single laboratory in Conrad and Johnson (2019). Those authors provided expanded uncertainties of 20 %, 44 %, and 17 % for EC, OC, and TC (from Table 2 in that work) relative to the 6.8 %, 13 %, and 4.7 % observed in the present work. Conrad and Johnson also determined that TC is the most repeatable, while OC is the least repeatable, again consistent with the current observations. The relative breakdown of within- and between-laboratory contributions to the uncertainties for TC here are also similar to the relative contributions observed by Schmid et al. (2001), though the uncertainties here are again smaller (expanded between-laboratory uncertainties of 18 % in Schmid et al. versus 12 % in this work). These collective observations indicate that the current measurements share many of the same trends as previous works but uncertainties in this work are consistently smaller. We hypothesize*

*that our smaller uncertainties are primarily due to the removal of volatile organics with a catalytic stripper, as organics are subject to transformation and mass loss during handling and storage. Since we observed lower uncertainties for TC than other studies, our lower uncertainties for EC and OC cannot be attributed to the EC/OC split alone. This is further supported by the lack of negative correlation between EC and OC (see Sec. 3.2), indicating that the split point was determined reliably. Further, it is likely that the use of a single particle source, a single thermal protocol, a single instrument model, and a common version of software all contribute to the smaller uncertainties observed in this study."*

*Similar discussion in the introduction has been updated.*

- Could the way each laboratories take their punches and/or their handling of the punches be responsible for the laboratory bias you reported (could one laboratory take their punch in a way that led to a systematic bias)?

  *While possible, this is highly unlikely. Randomization is implicit given that the loading was observed to be uniform and no identifying markers were present on the filter. We sought to note this in the manuscript by stating:*

  *"Punch positions on each sample were implicitly randomized by not otherwise providing further instruction to the laboratories. While this introduces a slight risk in the case of uneven filter loading, symmetry in the sampler and random filter orientations would minimize such risks in all but the center punch. The darkness of most filters was visually homogeneous, which further supports this decision."*

  *We have not updated this text.*

  *Note that if a laboratory takes their punch in a way that led to a systematic bias (such as rough handling that would cause loss of sample from the filter), it is important to capture this as part of the interlaboratory variability. We have added a note to this effect in the manuscript:*

  *"Further, even if there was a bias, for instance due to handling of the filter, it is important to capture this as part of the interlaboratory variability, as this would be representative of real world measurements."*

- I suggest adding more discussion on the implications and limitations of your work. By that, I mean discussing alternatives to TOA and how to reduce TOA uncertainty for regulatory aircraft nvPM mass emissions (use manual split, more thorough calibration procedures and quality controls, alternative calibration methods, etc).

*We have made attempts to better frame the impact and limitations of this work. We see limited scope for reducing the uncertainty with TOA through refinements to the calibration procedures and quality controls. One alternative to TOA that has emerged over the last decade is the CERMS (CPMA-Electrometer Reference Mass Standard), which has much lower uncertainties. While intermediate precision for CERMS has been reported as <3% (Ref. Titosky, 2019), to date there are no published results of an interlaboratory comparison with CERMS to provide comparable uncertainties. CERMS does appear to be a promising lower-uncertainty alternative to TOA for the calibration of nvPM mass concentration instruments.*

*A caveat has been added to the text:*

*"The authors see limited scope for reducing the uncertainties of TOA through refinements to the calibration procedures and quality controls. While promising alternatives to TOA are emerging for calibration of instruments, such as the CPMA-Electrometer Reference Mass Standard (CERMS) (Titosky et al., 2019; Corbin et al., 2020), the corresponding inter-laboratory variability of these alternatives have yet to be validated and should be a topic of future work."*

*Titosky, J., Momenimovahed, A., Corbin, J., Thomson, K., Smallwood, G., & Olfert, J. S. (2019). Repeatability and intermediate precision of a mass concentration calibration system. Aerosol Science and Technology, 53(6), 701–711.*

*Corbin, J. C., Moallemi, A., Liu, F., Gagné, S., Olfert, J. S., Smallwood, G. J., and Lobo, P.: Closure between particulate matter concentrations measured ex situ by thermal–optical analysis and in situ by the CPMA–electrometer reference mass system, Aerosol Science and Technology, 54, 1293-1309, 2020.*

---

## Author Comment (AC2)

Review of "Quantifying the uncertainties in thermal-optical analysis of carbonaceous aircraft engine emissions: An interlaboratory study"

This article presents an interesting and pertinent study on the estimation of measurement uncertainties on the total carbon (TC), elemental carbon (EC) and organic carbon (OC) contents, measured with the instrument Sunset 5L owned by five certified laboratories worldwide. Six equally shared punches from 20 samples probed from the exhaust of a helicopter engine were distributed for analysis among the chosen laboratories. Each instrument user respected the NIOSH5040 analysis protocol. The database was analyzed with statistical multilevel models to identify or predict the uncertainty bias among the samples.

The article is well-written and fits the topic of this journal. The obtained results are highly interesting for research topics such as nvPM emissions measurement protocols, atmospheric measurements, and aviation emissions, measurements, and protocols. There are a few arguably contrasting points that deserve to be put in a better light or clarified.

**General remarks to clarify**

1. The text does not clearly state when the samples were obtained. Is it the same work as Olfert et al. (2017) or another specific study? Please clarify this aspect.

   *The samples were collected on the same engine in the same facility as that reported by Olfert et al. However, they were collected as part of a separate emissions measurement campaign, conducted in Oct. 2016. Olfert et al. obtained their results as part of the MANTRA campaign, which was conducted in March 2015. We have clarified this by stating the following in the manuscript:*

   *Emissions were collected from the exhaust of a  helicopter turboshaft engine using a single point sample probe, in a subsequent study to MANTRA (reported by Olfert et al., 2017), on the same model of engine and in the same facility.*

2. Despite the significant experimental work in the referenced work Olfert et al., 2017, this article does not state which engine operating conditions were used for the obtained samples. It may not seem relevant to the authors, but why not have well-identified conditions in which nvPM was produced by the engine? I think this information is essential since the same operating conditions of the engine were used for three sets of samples loaded with 50, 100 and 250 $\mu g/m^3$ of soot, while the last sample loaded with 500 $\mu g/m^3$ of soot was obtained by increasing the RPM of the engine. It is well known that changing the engine's operating conditions will impact the structure and morphology of soot particles/nvPM. Isn't this contradictory with what the authors state in the paragraph from the introduction containing Lines 74 to 77? The reader can find additional information about the sampled particles on the filter by identifying the

operating conditions in the specified article if the work is common and even though the detailed statistical analysis did not identify any correlation between the filter loadings and uncertainties (lines 183-184).

> *All nvPM samples were collected at high power conditions for the Gnome engine. The samples loaded at mass concentration of 50, 100 and 250 µg/m³ of nvPM were obtained with the engine running at a steady 22,000 rpm. To produce the higher nvPM concentration required for the samples loaded at mass concentration of 500 µg/m³, the engine was operated at a steady 23,000 rpm. In both cases, the engine is at high power, and this modest adjustment to the engine's operating condition is not anticipated to impact the structure and morphology of the nvPM particles, as compared to reducing to a low power condition such as the 13,000 rpm reported in Olfert et al. (2017). This is supported by Saffaripour et al. (2020) which shows no significant change in the morphology of the particles from the same engine model between 21,000 and 22,000 rpm.*

> *The following has been added to the manuscript:*

> *"All nvPM samples were collected at high power conditions for the Gnome engine. All the samples loaded at mass concentration of 50, 100 and 250 µg/m³ of nvPM were obtained with the engine running at a steady 22,000 rpm. To produce the higher nvPM concentration required for the samples loaded at mass concentration of 500 µg/m³, the engine was operated at a steady 23,000 rpm. Saffaripour et al. (2020) demonstrates that there is no significant change in the morphology of the particles from the same engine model for such modest changes in the rotation speed."*

> *Saffaripour, M., Thomson, K. A., Smallwood, G. J., & Lobo, P. (2020). A review on the morphological properties of non-volatile particulate matter emissions from aircraft turbine engines. Journal of Aerosol Science, 139, 105467.*

3. The filter holder from Figure 1 contains two filter holders in series. Was the second filter analyzed for some residual TC content, as presented in the work of Corbin et al. (2020)?

> *The reviewer is correct in noting that two filters in series were used. The front filter is used to collect the sample for TC, EC, and OC analysis. It is known that quartz filters adsorb gas phase organic artifacts, and the second quartz filter is used to correct the OC and TC measurements from the front filter for the gas phase organics that were adsorbed on the front filter.*

> *The following has been added to the manuscript:*

> *"Quartz filters adsorb gas phase organic artifacts, and following the procedure outlined in Corbin et al. (2020), the data from TOA of the quartz filter in the second filter holder shown in Figure 1 is used to correct the OC and TC measurements from the front filter for the gas phase organics that were adsorbed on the front filter."*

4. It is surprising that the different loadings of the samples do not affect the uncertainty measurement of the three quantities measured by the Sunset instrument. This finding deserves a more detailed discussion since studies show that the loading of the filter impacts the uncertainty measurement of the thermo-optical analysis measurements.

> *The sampling times were adjusted such that the loadings were similar for all filters, regardless of the source concentration. Minor loading differences were observed, but they were small relative to differences in the mass concentration.*

> *The following has been added to the manuscript:*

> *"To compensate for the different mass concentrations used for loading the filters, the sampling time durations were adjusted such that the mass loadings were similar for all 20 filters."*

5. The use of the word structure (lines 183, 225, 229) and structural trends (line 227) can sometimes be misleading in the text for readers who are not specialized in multilevel statistical analysis. Please be more specific where it is the case; such as data/uncertainty structure or something that fits better in the context.

> *The terminology around "structured" errors has been removed from the manuscript in favor of discussion around systematic biases or effects. The biases also have some surrounding description to clarify the author's intention:*

> *"Results for EC and TC exhibit a consistent bias (or systematic error (JCGM, 2008)) across the different filters, where a laboratory that measured a value above average generally did so for all of the filters."*

**Specific comments:**

Figure 2 - to which sample corresponds to the obtained data? it is worth mentioning.

> *The data shown does not correspond to any particular measurement in this work. The caption has been updated accordingly:*

> *"Figure 2. Representative example of a TOA thermogram for nvPM emissions collected from the engine used in this study. Shown are the thermal protocol for aircraft engine emissions (SAE, 2018; Lobo et al., 2015a), the sample temperature, the FID signal, and the laser transmission measurement."*

Figure 6 - what represents the error bars in the bottom graphs with the Relative value [%] since it is mentioned that the error bars are excluded for clarity?

*The error bars correspond to laboratory-reported uncertainties on individual points. The caption has been updated:*

*"Error bars in the lower panels correspond to expanded (k = 2) uncertainties reported by the laboratories and, while only included for select points, were similar across all of the data."*

Line 37: remove on from the sentence "... mass on collected ..."

*This change has been made as recommended.*

Line 110: "darkness ..." can be replaced with "coverage ..." to differentiate from dark uncertainty

*Coverage is now used in connection with expanded (k = 2) uncertainties. We rather use "loading":*

*"The loading of most filters was visually homogeneous, which further supports this decision."*

Line 136: remove the from the sentence "... and the their uncertainty ..."

*This sentence was removed amongst the other changes.*

Line 199: The authors mentioned, "These filters coincide with cases where the overall variance is larger and represent a minority of cases." Please be more specific when selecting a criterion for the value of the variance to eliminate the sample in question. Either be specific and justify why this selection was made or mention if you referred to data outliers.

*The data for these samples were not excluded from the analysis, such that no specific criterion was applied to remove the data or identify them as outliers. Further, the updated statistical model also no longer relies on laboratory-reported uncertainties to determine within-laboratory variation, such that these variances are less relevant to the analysis (only introduced as weak prior information). Overall, the quoted sentence added more confusion than clarification and has thus been removed from the revised manuscript.*